# Mechanisms of Regulation of the Expression of miRNAs and lncRNAs by Metformin in Ovarian Cancer

**DOI:** 10.3390/ph16111515

**Published:** 2023-10-24

**Authors:** Ignacio Alfaro, Margarita Vega, Carmen Romero, Maritza P. Garrido

**Affiliations:** 1Laboratory of Endocrinology and Reproductive Biology, Clinical Hospital University of Chile, Independencia 8380453, Chile; 2Obstetrics and Gynecology Department, Faculty of Medicine, University of Chile, Santiago 8380453, Chile

**Keywords:** metformin, microRNAs, long non-coding RNAs, ovarian cancer

## Abstract

Ovarian cancer (OC) is one of the most lethal gynecological malignancies. The use of biological compounds such as non-coding RNAs (ncRNAs) is being considered as a therapeutic option to improve or complement current treatments since the deregulation of ncRNAs has been implicated in the pathogenesis and progression of OC. Old drugs with antitumoral properties have also been studied in the context of cancer, although their antitumor mechanisms are not fully clear. For instance, the antidiabetic drug metformin has shown pleiotropic effects in several in vitro models of cancer, including OC. Interestingly, metformin has been reported to regulate ncRNAs, which could explain its diverse effects on tumor cells. In this review, we discuss the mechanism of epigenetic regulation described for metformin, with a focus on the evidence of metformin-dependent microRNA (miRNAs) and long non-coding RNA (lncRNAs) regulation in OC.

## 1. Introduction

### 1.1. Ovarian Cancer

Ovarian cancer (OC) is a lethal gynecological malignancy. According to the GLOBOCAN 2020 statistics, there were 314,000 new cases and 207,000 deaths in the world, projecting a 100% increase in OC deaths by 2040 [1]. Epithelial OC (EOC) is the most common subtype of OCs (representing over 95% of OC) and is characterized by non-specific symptoms and late diagnosis, which results in poor prognosis. High-grade serous epithelial OC (HGS-EOC) is very aggressive and represents 75% of OC [2,3].

From a molecular point of view, inflammation and angiogenesis (the formation of new blood vessels from pre-existing ones) are two important and connected processes involved in the progression, chemoresistance, and dissemination of OC [4,5]. Chronic inflammation is strictly related to cancer risk, producing an increase in cell proliferation and reduced DNA repair [6]. Both extrinsic and intrinsic inflammation (produced by the immune system and by the own cancer cells, respectively) contribute to multiple hallmark capabilities by supplying bioactive molecules to the tumor microenvironment, including growth factors that sustain proliferative signaling, survival factors that limit cell death, proangiogenic factors, extracellular-matrix-modifying enzymes that facilitate angiogenesis, invasion, and metastasis, and inductive signals that lead to the activation of epithelial-mesenchymal transition (EMT) [7]. In EOC, increased expression of cyclooxygenase (COX)-1 and COX-2, and the increase in inflammatory mediators, such as prostaglandin E2 (PGE2), have been associated with poor prognosis [8,9,10]. The inflammatory signaling promotes the survival of cancer cells and causes genomic instability, allowing mutant cells to escape cell cycle arrest and apoptosis [11].

Another key characteristic of EOC is its high angiogenesis potential, since EOC cells produce several angiogenic factors, such as the vascular endothelial growth factor (VEGF) family, angiopoietins, neurotrophins, fibroblast growth factors (FGF), and platelet-derived growth factor (PDGF), among others [12,13]. This knowledge has served to develop several anti-angiogenic therapies that are being tested in patients with OC [12], including Bevacizumab, a monoclonal antibody against VEGF-A that was approved by the FDA in 2018 for use in patients with advanced-stage OC.

### 1.2. Current and New Approaches for OC

The current treatment for EOC is debulking surgery. Primary cytoreductive surgery followed by adjuvant chemotherapy remains the standard treatment for EOC in advanced stages [14]. Despite optimal surgery and chemotherapy, ∼70–80% of patients with EOC will develop disease relapse [14,15], indicating the need to improve the existing therapies.

New therapeutic alternatives have been studied for OC; some of them are natural compounds, such as flavonoids and polyphenols, which have shown antitumoral effects in several kinds of cancer, including OC [16]. Other natural compounds, such as withanolides (bioactive molecules isolated from *Whitania somnifera* or Indian ginseng), have shown enhanced therapeutic activity as anticancer drugs, suppressing growth and metastasis in OC models [17,18]. In this line, the design of small peptides that inhibit mutant oncogenic proteins, such as K-Ras, one of the major gene mutations correlated with OC occurrence [19], has promising results [20]. On the other hand, the use of repurposing drugs (or the investigation of approved drugs for new therapeutic purposes, such as cancer treatment) is being considered as a therapeutic approach in OC [14]. Thus, drugs such as metformin have been studied in in vitro and in vivo models and are currently being tested in several clinical trials as a complementary therapy for OC.

### 1.3. Metformin in OC

Evidence from retrospective studies has shown that the use of some drugs, such as metformin, could decrease OC incidence and mortality in diabetic patients [21]. Metformin is a widely used drug for the treatment of metabolic disorders, such as type 2 diabetes mellitus, metabolic syndrome, gestational diabetes, and polycystic ovarian syndrome [22,23]. Metformin produces normoglycemia by decreasing hepatic glucose production, the intestinal absorption of glucose, and improving insulin sensitivity, thus promoting glucose uptake and utilization by peripheral tissues [22]. Metformin enters the cell through organic cation transporters (OCTs) and multidrug and toxin extrusion transporters [24,25] and is accumulated in tissues that express OCT transporters, such as the ovary [26,27,28], which are, therefore, adequate targets for metformin action. In the context of cancer, the direct antitumoral effects of metformin are pleiotropic and involve several molecular targets at different levels in the tumoral cell, including epigenetic regulation (changes in cell function without alterations in the DNA sequence) [29].

In vitro experiments showed that metformin decreased EOC cell proliferation by inducing cell cycle arrest and altering glucose and lipid metabolism [30,31]. In vivo experiments showed that metformin treatment decreased OC growth and the presence of cell proliferation markers (such as Ki-67 and cyclin D1), as well as metastasis [32,33]. Most of these antitumoral effects exerted by metformin depended on the activation of the enzyme 5′ adenosine monophosphate-activated protein kinase (AMPK), which induces p53 phosphorylation (S15), which is required for cell cycle arrest [34]. In addition, high doses of metformin inhibited the mitochondrial respiratory chain complex I, activating AMPK, increasing reactive oxygen species (ROS), and producing a glycolytic switch in OC cells [35].

In addition to its anti-proliferative effects, metformin treatment inhibits both angiogenesis and metastatic spread in OC models [32,33]. In vitro and in vivo studies showed that mechanistic inhibition of angiogenesis by metformin involves the inhibition of the expression of angiogenic factors, such as VEGF, as well as a decrease in endothelial cell migration [36] and the polarization of tumor-associated macrophages (TAMs) [37].

On the other hand, metformin has been described to prevent the EMT phenotype in vitro [38], which is associated with cell cycle arrest and the accumulation of OC cells in the S-phase of the cell cycle [39]. In agreement with the last point, several in vitro and in vivo studies have shown that platinum or taxane therapy allows OC cells to acquire a mesenchymal phenotype [40,41,42]. EMT proteins such as Snail, Slug, Twist-1, and Zeb-1 contribute not only to OC dissemination but also increase cisplatin and carboplatin resistance in OC cells [41,43,44,45]. In addition to these antecedents, evidence suggests that metformin targets cancer stem cells: a phase II clinical trial showed that metformin decreased the population of aldehyde dehydrogenase (ALDH)(+) and CD33(+) cells in ovarian tumors [46]. All of this evidence could explain the inhibitory effects of metformin on the metastatic behavior of OC cells.

Metformin has pleiotropic effects and targets OC cells at multiple levels. Since non-coding RNAs (ncRNAs) can regulate several proteins and produce multiple changes at cellular levels, it is plausible that the antitumoral effects of metformin in OC could be explained by ncRNA regulation. In fact, recent evidence has shown that metformin treatment could regulate the expression of ncRNAs such as microRNAs (miRNAs) and long non-coding RNAs (lncRNAs) to exert antitumoral effects.

Few studies have shown that the antitumor effects of metformin can be attributed directly, or at least in part, to the modulation of ncRNAs in OC. In this review, we discuss the existing evidence regarding how metformin could modulate the levels of some important ncRNAs in OC. Since the studies are limited, we decided to discuss a possible connection between the changes in some ncRNAs in OC progression and their modulation by metformin, with a focus on miRNAs and lncRNAs.

## 2. Methodological Approach: Search Strategy and Articles Inclusion Criteria

We searched the PubMed database for articles that described the effects of metformin on cancer cells and whose antitumoral mechanisms are related to the modulation of ncRNAs. Articles were published between 2006 and 2023. The following search terms were used: (“metformin”) AND (“cancer” OR “carcinoma”) AND ((“noncoding RNAA”; OR “ncRNA”) OR (“microRNA” OR “miR”) OR (“long noncoding RNA”; OR “lncRNA”) OR (“circRNA” OR “circular RNA”)). A total of 255 potential studies were found through database searching, and an initial screening was performed, including only articles written in English.

The PubMed database was also used to search for articles describing relevant oncogenic ncRNAs in OC that are regulated by metformin, as indicated in the former articles found. The following search terms were used: “miR-21” AND (“ovarian cancer”; OR “ovarian carcinoma”); “miR-27a” AND (“ovarian cancer”; OR “ovarian carcinoma”); “H19” (long non-coding RNA H19) AND (“ovarian cancer”; OR “ovarian carcinoma”); “SNHG7” (mall Nucleolar RNA Host Gene 7) AND (“ovarian cancer”; OR “ovarian carcinoma”). The number of articles found related to relevant oncogenic ncRNAs in OC was 88 (miR-21), 22 (miR-27a), 39 (H19), and 4 (SNHG7).

To discuss and propose the new ncRNAs that metformin could potentially regulate in OC, we considered those articles from the initial screening that explored modulated ncRNAs by metformin in cancer cells, including OC cells, according to the following criteria: (1) the articles describe metformin-regulated ncRNAs that are considered oncogenic in OC, and (2) ncRNAs that are downregulated by metformin. We further considered these oncogenic ncRNAs according to their relevance in OC development and progression, and the number of articles that resulted from the PubMed database search.

## 3. Non-Coding RNAs

Non-coding RNAs (ncRNAs) play a key role in the pathophysiology of many diseases, including cancer. There are different kinds of ncRNAs: transcripts of less than 200 nt are small non-coding RNAs, while transcripts bigger than 200 nt are lncRNAs, of which miRNAs are the main class of ncRNAs [47]. The regulation of the expression of ncRNAs by metformin may involve epigenetic modifications (for instance, modification of DNA methylation) as well as transcription regulation and modulation of maturation.

Figure 1 shows the diagram of miRNA synthesis, which begins with the transcription of a primary miRNA (pri-miRNA). Then, an imperfect hairpin structure is cleaved, producing a smaller hairpin called precursor miRNA (Pre-miRNA). It is exported from the nucleus to the cytoplasm through exportin 5, a double-stranded RNA-binding protein. The Pre-miRNA is processed by the RNase III Dicer, a double-stranded RNA nuclease, producing a small miRNA duplex. In the next step, the miRNA duplex (miR duplex) binds Argonaute (AGO) proteins, a family of gene-silencing proteins guided by small RNAs, such as miRNA, that get into specific binding pockets and guide AGO proteins to target messenger RNAs (mRNA) [48]. After AGO ejects the passenger strand (a strand that will not be used to form the silencing complex), the guide strand bound to AGO, along with other proteins, will form the RNA-induced silencing complex (RISC), which will produce the degradation of the mRNA of complementary sequence to the miRNA or the inhibition of its translation [49].

## 4. Mechanisms of ncRNA Biosynthesis Regulated by Metformin

The pleiotropic anticancer effects of metformin have been studied recently through multiple pathways, including its ability to regulate the expression of ncRNAs [50]. Metformin modulates the expression of several ncRNAs in different cancer cells, including OC cells, affecting their tumor formation and progression capacity [51]. However, most studies have focused on describing changes in ncRNA levels upon metformin treatment, and very few have explored the regulatory mechanism by which metformin alters the expression of these ncRNAs. To help build a more in-depth evidence framework, in this section, we discuss the previously described mechanisms through which metformin manages to alter the expression of ncRNAs, with a focus on OC and miR-145.

### 4.1. Role of Transcriptional Factors in the Metformin-Mediated Modulation of miRNA Expression

#### p53-Dependent Metformin Effects

The p53 protein is an important and well-studied tumor suppressor transcription factor. It can regulate the expression of miRNAs either through transcriptional-dependent modulation and/or post-transcriptional processing and maturation to induce tumor suppressor biogenesis of miRNAs in several human cancers [52]. Metformin’s ability to regulate transcription factors to modulate miRNA expression has been studied in cancer cells. Two reports have shown that increasing levels of miR-34a and miR-23a tumor suppressors by metformin treatment require a concomitant induction of p53 in breast cancer and hepatocellular carcinoma cells, respectively [53,54]. Neither of these miRNAs were induced by metformin in p53-deficient or p53-mutant cells, suggesting that wild-type p53 is necessary for metformin to induce tumor suppressor miRNA. Notably, miR-145 is a p53 transcriptional target, and additionally, the direct interaction of p53 with Drosha promotes miR-145 pri-miRNA processing into its mature and active form in colorectal and breast cancer cells [55,56]. Studies conducted in the A2780 EOC cell line, which harbors wild-type p53 [57], have shown that metformin increases p53 protein levels [58] and that doxorubicin (a p53 activator) treatment causes upregulation of miR-145 levels [59].

Unfortunately, most human malignancies possess mutations in the *TP53* genes, including high-grade serous carcinomas, which often involve p53 mutations [60]. As reported, mutants of the p53 protein can exert opposite effects to its wild-type counterpart; that is, disrupt the processing of pri-miRNAs, lowering the maturation of miR-145 in colorectal cancer cells [56]. Interestingly, metformin has been shown to downregulate p53 expression in OV90 cells (EOC cells that carry a mutated *TP53* gene [61]). Considering that mutant p53 downregulates miR-145, a metformin-mediated decrease in mutant p53 levels could possibly lead to an induction of miR-145 in EOC cells. However, studying the effects of metformin on miR-145 in this type of cancer cell is still required to prove this mechanism. In summary, wild-type p53 is necessary for the induction of tumor suppressor miRNAs (such as miR-34a and miR-23a) by metformin in breast cancer and hepatocellular carcinoma cells [51,62]. In the case of OC cells, metformin possibly increases miR-145 levels through the upregulation of wild-type p53 or downregulation of mutant p53, highlighting the potential versatility of metformin in terms of p53 regulation to positively alter miR-145 levels, whereas miR-145 induction in p53-lacking cells could also be a possibility.

The E2F transcription factor 3 (E2F3) has also been shown to play a role in the metformin-mediated modulation of miRNA expression. E2F3 belongs to the E2F family of transcription factors that have crucial roles in regulating cell cycle progression and tumorigenesis [63]. E2F3 can have both activating and repressing transcriptional activity [64]. Metformin negatively regulates oncogenic miR-21 by increasing the expression of E2F3, favoring the repression of transcription at the miR-21 promoter in breast cancer cells [65]. In addition, another study carried out by the same authors, conducted in a similar experimental context, revealed that metformin treatment increases the activity of E2F3 on the DICER promoter, increasing its expression [66]. In consequence, Dicer-sensitive miRNAs, such as tumor suppressor miR-33a, were upregulated by metformin treatment. The authors of these two reports discuss that their results indicate that both effects mediated by metformin may be independent of each other and could reflect a tumor- and stage-specific mechanism. Thus, these results suggest that metformin can influence the activity of E2F3 to repress oncogenic miRNA expression and/or upregulate Dicer expression to promote tumor suppressor miRNA processing and maturation.

As reported, E2F3 is overexpressed [67], and miR-145 is downregulated [68] in human OC tissues. In addition, E2F3 was shown to be a direct target of miR-145 in EOC cells [69]. Accumulating evidence has described that E2F3 and miRNAs form negative feedback loops, where their expression is mutually influenced [64]. Therefore, this kind of relationship could exist for E2F3 and miR-145 in EOC cells, thus impairing miR-145 expression. Since metformin can influence E2F3 activity and upregulate miR-145 levels, it could possibly modulate the E2F3/miR-145 ratio in OC to produce a net increase in miR-145 levels. However, the role of E2F3 in metformin-mediated regulation of miRNAs, such as miR-145, remains to be elucidated in EOC.

### 4.2. Role of Metformin on Epigenetic Modification of ncRNAs

#### 4.2.1. DNA Methylation of ncRNAs

Epigenetic regulation involves changes in cell function without alterations in the DNA sequence. It includes DNA methylation, histone modifications, and post-transcriptional gene regulation by ncRNAs [70].

Metformin downregulates the oncogenic lncRNA H19 (H19) in OC cells by increasing the methylation level in the H19 promoter region [71]. Additionally, H19 acts as a molecular sponge to sequester let-7 [72], a miRNA family that acts as a tumor suppressor in EOC [73]. As reported, metformin-mediated downregulation of H19 frees let-7 from sequestration, thereby increasing its bioavailability in OC cells [71]. A similar effect was observed in hypopharyngeal cancer cells, where metformin increased the methylation in the promoter of the oncogenic lncRNA SNHG7 to repress its expression [74]. Consistently, metformin was also shown to downregulate SNHG7 expression in EOC cells [75]. Notably, as observed in the H19/let-7 interaction, downregulation of SNHG7 releases miR-3127 (which acts as a tumor suppressor) from its sequestration, increasing its expression and bioavailability. Therefore, metformin downregulates oncogenic lncRNAs by inducing hypermethylation of their promoters, and due to this effect, it also favors the release of tumor suppressor miRNAs sequestered by lncRNAs, increasing their bioavailability.

However, a contrasting effect of metformin was observed on the methylation status of miRNA genes. In pancreatic cancer cells treated with metformin, the promoter of the tumor suppressor miR-663 was hypomethylated, and its expression was increased [76]. Another study showed that metformin increased tumor suppressor miR-570 by DNA demethylation of its promoter in osteosarcoma cells [77]. Moreover, a strong body of evidence suggests that the promoter of the miR-145 gene is highly hypermethylated and that inhibitors of DNA methylation and DNA methyltransferases (DNMTs) are able to increase miR-145 expression in EOC cells [78,79]. Given all of this background regarding how metformin upregulates the expression of miR-145 in EOC cells [80], it could achieve this effect through epigenetic-related mechanisms that lead to hypomethylation of the miR-145 gene promoter. However, this idea has not been studied yet.

The mechanisms by which metformin alters the methylation state of ncRNA genes have been related to the ability of metformin to regulate the activity of S-adenosylmethionine (SAM)-dependent DNMTs. Metformin has been shown to decrease intracellular S-adenosylhomocysteine (SAH), a strong feedback inhibitor of SAM-dependent DNMTs, by promoting the enzymatic activity of SAH hydrolase (SAHH) in non-cancerous cells and breast, endometrial, and hypopharyngeal cancer cells [74,81,82]. Metformin can also modulate the expression of DNMTs, downregulating their expression in lung cancer cells [83] and, conversely, upregulating their expression in hypopharyngeal cancer cells [74]. Therefore, the influence of metformin on DNMTs and its effects on the methylation status of ncRNAs appear to be tumor-dependent. Additionally, it seems that metformin could “select” ncRNA genes for epigenetic regulation since it specifically hypermethylates oncogenic ncRNAs and hypomethylates ncRNAs. Although this behavior of metformin is consistent with its proposed anticancer mechanism of action, the relationship between methylation and gene expression is more complex, and the widely used assumption that hypermethylation results in gene expression inhibition (and vice versa) may not always be accurate.

#### 4.2.2. Methylation of miRNAs

Metformin can also regulate the expression of miRNAs through mechanisms involving N6-methyladenosine (m6A) methylation. The role of m6A modifications in ncRNAs is a new topic of research in cancer that has been extensively reviewed by Ma et al. [84]. One study reported that metformin decreased the activity of DNMTs on the methyltransferase-like 3 (METTL3) promoter (an enzyme that causes m6A modifications in RNA), decreasing its methylation and favoring its expression and activity to promote let-7b expression in lung cancer cells [85]. Mechanistically, METTL3 methylates pri-let7b, allowing its recognition by NF-kappa-B-activating protein (NKAP) and heterogeneous nuclear ribonucleoproteins A2/B1 (HNRNPA2B1), which assist the Drosha-DGCR8 complex in processing pri-let7b into mature let-7b. Overall, the evidence shows that metformin can promote the m6A modification of primary miRNAs, such as let-7b, to favor their maturation.

In OC, METTL3 has been considered an oncogenic gene [86]. Importantly, metformin has been shown to decrease METTL3 in a mouse breast cancer model [87]. Thus, it is possible that metformin could alter METTL3 expression in OC to induce tumor suppressor miRNAs, such as let-7, by an m6A modification-associated mechanism. However, the findings on the role of m6A modification in ncRNAs are recent, and further studies are necessary to understand how metformin actually exerts this effect.

### 4.3. Role of Metformin in miRNA Maturation

Several studies have suggested that post-transcriptional maturation, rather than transcription, is often altered in cancer [88]. Accordingly, pri-miRNAs accumulate and deplete mature miRNAs in human cancer [89], indicating that the machinery for processing and maturation of miRNAs is dysregulated in cancer. Notably, metformin has been shown to induce the expression of key processing enzymes of miRNA biogenesis. Metformin treatment upregulates Drosha and Dicer expression in cholangiocarcinoma and breast cancer cells, respectively, to modulate miRNA expression [66,90]. On the other hand, decreased Dicer and Drosha levels represent an oncogenic event in EOC cells and are associated with poor patient outcomes [91,92,93]. Interestingly, p70S6 kinase (p70S6K), a downstream effector of phosphoinositide 3-kinases (PI3K)/protein kinase B (Akt)/mechanistic target of rapamycin (mTOR) signaling, was shown to affect the miRNA biogenesis machinery in EOC cells [94]. P70S6K phosphorylates tristetraprolin (TTP), preventing its interaction with Dicer, which specifically promotes the maturation of miR-145. Metformin may also regulate miRNA expression by influencing the activity of miRNA processing enzymes in OC. As reported, metformin impairs PI3K/Akt/mTOR signaling and thus decreases the active phosphorylated form of P70S6K in EOC cells [58,95,96]. Since metformin can increase miR-145 levels in EOC cells [80], these findings suggest that metformin may prevent p70S6K activation to favor miR-145 expression through Dicer and TTP interaction in EOC cells.

## 5. ncRNA-Related Therapeutic Effects of Metformin in OC

Very few studies have reported that metformin’s anticancer effects may involve the regulation of ncRNAs in OC. Together with the studies that our group has carried out regarding the regulation of tumor suppressors miR-145 and miR-23b by metformin [80], only two additional studies reported an influence of metformin on ncRNAs in OC [71,75]. These studies showed that treatment of OC cells with metformin reduced the expression of the lncRNAs H19 and SNHG7, which was associated with anticancer effects. Because there are few studies related to this topic in OC, in this section we describe some important ncRNAs in the pathogenesis and progression of OC and discuss their potential as relevant metformin targets.

### 5.1. miR-23b and miR-145 in OC

miR-23b and miR-145 are two oncosupressor miRNAs that are downregulated in EOC cell lines and EOC tissues, as we have reported [97]. To date, our group has shown that metformin increases the expression levels of tumor suppressors miR-145 and miR-23b in EOC cells and prevents the nerve growth factor (NGF)-induced decrease in these miRNAs [80]. These changes coincided temporarily with a decrease in the expression and transcriptional activity of c-MYC upon metformin treatment. Other authors have shown that c-MYC can reduce the expression of both miR-145 and miR-23b in cancer [98,99]. Furthermore, a study performed with a small sample of EOC patients showed that metformin intake reduced the presence of oncoproteins related to cell proliferation, such as c-MYC and survivin [80]. Consistent with these findings, we have further shown that NGF reduces the transcriptional activation of the miR-145 promoter while inducing c-MYC protein levels in EOC cells [68]. Therefore, NGF could repress miR-145 and miR-23b, and these effects could be mediated by c-MYC activation, while in the presence of metformin, this NGF ability is impaired. These observations suggest that metformin potentially modulates transcription factors, such as c-MYC, to increase tumor suppressor miRNA levels in OC cells.

### 5.2. miR-21 in OC and Other Cancers

miR-21 is considered a pro-tumoral miRNA in EOC since its overexpression increases cell proliferation, invasion, and migration abilities of EOC, and decreases apoptotic cell death in EOC tumors [100,101,102,103]. Additionally, miR-21-5p is involved in paclitaxel resistance in EOC cells because it can sensitize OC cells to paclitaxel, reducing cell proliferation, migration, invasion, and EMT [104].

Although there is vast evidence of the pro-tumoral role of miR-21 in EOC, and metformin has been shown to downregulate this miRNA in several kinds of cancer, the direct effect of metformin on miR-21 in EOC has not been studied yet. Nevertheless, a connection must exist between metformin and miR-21 since metformin downregulates this miRNA in a diversity of human cancer models. For instance, a reduction of miR-21 levels in breast cancer cells following metformin treatment has been assessed both in vitro and in vivo [65,105]. Also, in chemo-resistant colon cancer cells (highly enriched in cancer stem cells), metformin causes a marked reduction of miR-21 expression, and this effect is further accomplished in combination with oxaliplatin treatment [106]. Metformin has been shown to act synergistically with oxaliplatin to induce cell death, inhibit colonosphere formation and cell migration, and inhibit tumor growth of colon cancer cells. In the case of renal cancer cells, metformin treatment can decrease miR-21 expression and increase PTEN levels, impairing PI3K/Akt signaling and cell proliferation [107]. A similar effect has been shown in hypopharyngeal cancer cells, in which metformin treatment inhibited miR-21 expression, causing a decrease in cell proliferation [54].

Additionally, metformin has shown anti-angiogenic effects in EOC cells; it improves OC sensitivity to paclitaxel and platinum-based drugs and decreases the metastatic potential of OC cells [31,80,108]. Since the upregulation of miR-21 has been described in these processes [109,110], it is possible that these effects could be mediated by a metformin-dependent decrease in miR-21.

### 5.3. miR-27 in OC and Other Cancers

Another important miRNA is miR-27a, which represents an important oncomiR involved in OC development, progression, and chemoresistance. It stimulates cell proliferation, cell cycle progression, migration, invasion, and EMT of EOC cells by directly targeting a tumor suppressor transcription factor, forkhead box O1 (FOXO1). Metformin has been shown to downregulate miR-27a in cancer cells of several origins. For example, in breast cancer cells, metformin inhibits growth and promotes apoptosis by decreasing miR-27a levels [111]. Additionally, the expression of miR-27a is upregulated in taxol-resistant EOC cells, which in turn could increase multidrug-resistant 1 (MDR1) expression. Since miR-27a could directly target AMPK [111], metformin could activate AMPK by downregulating miR-27a in cancer cells. On the other hand, miR-27a levels decreased upon metformin treatment in pancreatic cancer cells, upregulating transcription factor repressors (zinc finger and BTB domain-containing protein 10 (ZBTB10)) [112] of the specificity protein (Sp). This leads to downregulation of Sp1/3/4 transcription factor and several pro-angiogenic Sp-regulated genes, including bcl-2, survivin, cyclin D1, VEGF, VEGF receptor, and fatty acid synthase, which may account for the antitumoral effects of metformin in pancreatic tumor growth. The effect of metformin on miR-27a has not been assessed yet in OC. Since miR-27a is also an important oncomiR that has overlapping target genes with miR-21, its downregulation could explain in part the anti-angiogenic, anti-metastatic, and chemo-sensitizing effects of metformin in OC.

In summary, solid scientific evidence has shown that metformin can regulate miRNA abundance, such as miR-145 and miR-23b, in EOC. On the other hand, metformin regulates miR27a and miR21 in different kinds of cancer, suggesting that metformin could be implicated in the regulation of these miRNAs in OC. However, metformin not only modulates the expression and abundance of miRNAs but also lncRNAs (Table 1 and Figure 2). In fact, a wide variety of these ncRNAs can be regulated by metformin. Next, we will review the evidence of the effects of metformin on two very important lncRNAs in EOC: H19 and SNHG7.

### 5.4. lncRNA H19 in EOC and Its Regulation by Metformin

H19 is a lncRNA widely considered an oncogene in various types of cancer, including EOC, and it is critically involved in tumor development, malignant progression, and chemoresistance [125]. H19 is highly expressed in human ovarian tumor tissues and has been associated with cancer progression and poor patient prognosis [126,127]. Moreover, high levels of H19 can also be detected in ascite fluids from OC patients [128], suggesting a role in metastasis development. Knockdown of H19 has been shown to decrease cell proliferation, migration, and invasion in OC cells [129,130,131]. In addition, H19 acts as a molecular sponge of miRNAs [127,132], and controls OC metabolism (favoring the Warburg effect) [133]. Moreover, H19 also participates in the development of chemoresistance in EOC. H19 can enhance the chemoresistance of OC cells to carboplatin by antagonizing miR-29, which increases multidrug resistance proteins, such as MDR1 and MRP1 [134]. LncRNA H19 contributes to enhancing the growth and cell cycle of cancers and inducing EMT and, therefore, promotes metastasis [135]. Importantly, the knockdown of H19 in cisplatin-resistant EOC cells improves cisplatin sensitivity in vitro and in vivo through glutathione metabolism impairment [136]. This report indicated that H19 is involved in tumor development, malignant progression, and resistance to chemotherapy in EOC, and, therefore, it profiles as a relevant therapeutic target.

In terms of how metformin regulates H19 in EOC cells, Yan et al. demonstrated that metformin decreases H19 expression by altering its promoter methylation, inhibiting EOC cell migration and invasion [71]. Accordingly, it was previously shown that histone H1.3 directly represses the expression of the H19 gene in EOC cells through a similar epigenetic mechanism involving DNA methylation [137]. Downregulation of H19 relieves let-7 inhibition from H19 sponging, leading to decreased metastasis-promoting genes such as c-MYC, the high-mobility group AT-hook 2 (HMGA2), and U3 small nucleolar ribonucleoprotein (IMP3) [71]. Although evidence suggests that the use of metformin might be useful in targeting H19 in OC therapy, these findings are limited to cell lines, and further studies are needed to test this metformin effect in vivo. Nevertheless, according to the above, studies in other cancer types have supported the idea that metformin could be clinically useful in targeting H19 and producing antitumor effects. For instance, endometrial cancer patients receiving antidiabetic doses of metformin have reduced H19 expression levels in the endometrial tumor tissue [82,120]. In preclinical studies, metformin treatment has been shown to inhibit cell migration and invasion by downregulating H19 expression in gastric cancer cells [121]. Furthermore, in a mouse model injected intravenously with gastric cancer cells, metformin suppressed metastasis in a similar way to that of mice that were injected with H19-knockdown cells. Similarly, a recent study conducted in breast cancer cells showed that metformin may induce ferroptosis by downregulating H19 [116]. All this evidence shows that metformin decreases H19 in different cancers, and the pivotal role of H19 in EOC supports further metformin research as a relevant H19-targeting therapeutic agent in OC.

### 5.5. Metformin Regulation of lncRNA SNHG7 in OC

The lncRNA SNHG7, along with the lncRNA H19, is one of the well-studied oncogenes involved in the development of multiple cancers [138]. However, its study in EOC is rather scarce. In ovarian tumor tissues, SNHG7 expression was shown to be elevated [139], suggesting its role as an oncogene. Consistently, the knockdown of SNHG7 in OC cells decreased their growth, migration, and invasion abilities and inhibited tumorigenesis in vivo [139]. Additionally, silencing SNHG7 expression enhanced the sensitivity of resistant OC cells to paclitaxel and reduced their migrative and invasive potential [140].

A recent study reported for the first time that metformin enhances paclitaxel sensitivity and inhibits cell viability, migration, and invasion by decreasing SNHG7 expression in paclitaxel-resistant OC cells [75]. Simultaneously, metformin was shown to promote miR-3127 expression, which proved to be a direct target of SNHG7, which acts as a molecular sponge. Importantly, the regulatory effect of metformin on the SNHG7/miR-3127 ratio was confirmed in a xenograft model of OC, in which metformin prevented the promotion of tumor growth by SNHG7 overexpression [75]. In addition, metformin can also downregulate SNHG7 expression in hypopharyngeal cancer cells to inhibit cell proliferation and improve paclitaxel sensitivity [141]. There is still very little evidence of the SNHG7 oncogenic role and its regulation by metformin as an anticancer mechanism in OC. However, we hope that the existing results will encourage further research as metformin increasingly continues to profile as an attractive therapeutic alternative in OC.

## 6. Future Directions

The antitumoral effects of metformin, as evidenced in in vitro and retrospective studies, have supported the study of metformin in non-diabetic patients with different kinds of cancer. There are more than 400 clinical trials registered in ClinicalTrials.gov that have tested metformin in patients with and without diabetes with different kinds of cancer. Several of these trials yielded results; however, the results are contradictory. For instance, in patients with breast cancer cells, the use of metformin vs. placebo did not significantly improve invasive disease-free survival in the entire arm. However, ERBB2+ patients had longer invasive disease-free survival [142]. Similarly, a study of patients with advanced pancreatic cancer showed that metformin does not improve outcomes in the group treated with gemcitabine, erlotinib, and metformin, but the overall survival was significantly longer in the 16 patients with higher concentrations of metformin [143]. In this line, a study performed on patients with breast cancer shows two different patterns: metformin responders and non-responders, suggesting that patients with an increase in oxidative phosphorylation gene transcription could be resistant to metformin treatment [144].

However, there are some pilot studies that tested the effects of metformin in patients without diabetes or metabolic abnormalities in a “preoperative window” or in patients with pre-malignant lesions, which show more consistent results. Table 2 summarizes some of these studies and their main findings.

Taking this into consideration, the different metformin responses observed in several studies could be a consequence of the different metformin concentrations allowed in blood or tissue, the presence of specific mutations, or a metabolic signature. We think that a miRNA profile could be useful to characterize patients who may respond better to metformin treatment. Even though almost every clinical trial that tested metformin in cancer patients did not consider the study of the miRNA profile (enrolled in ClinicalTrials.gov), some of them attempted to measure changes in miRNAs, which are listed in Table 3. The information provided by these studies can be very valuable in clarifying the differential response observed in cancer patients treated with metformin.

## 7. Conclusions

Increasing scientific and clinical evidence supports the antitumor effects of metformin in cancer, including OC. Several of its antitumoral effects can be explained as multiple effects at the post-transcriptional level, which include the modulation of miRNAs and lncRNAs studied in different models of cancer. Some possible mechanisms that could be implicated in the antitumoral effects of metformin in OC are the upregulation of miR-145 and miR-23b and the downregulation of miR-21 and miR-27a. Furthermore, metformin could downregulate some lncRNAs, such as H19 and SNHG7, which in turn could modulate miRNA expression and many protein targets of metformin. However, the evidence for the regulation of ncRNAs by metformin in OC is still scarce and needs to be further investigated.

## Figures and Tables

**Figure 1 pharmaceuticals-16-01515-f001:**
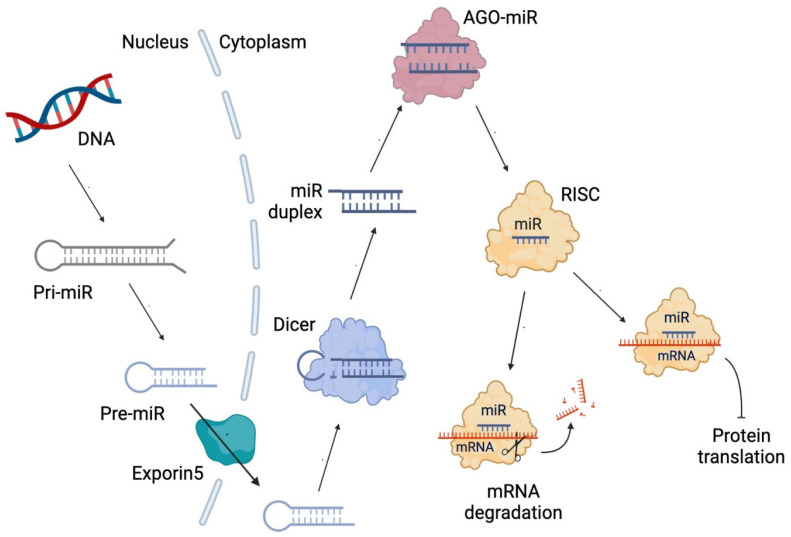
Diagram of microRNA (miRNA) biosynthesis. miR duplex: miRNA duplex comprising two strands. pri-miRNA: primary miRNA. Pre-miRNA: precursor miR. mRNA: messenger RNA. Dicer: RNase III double-stranded RNA nuclease. AGO-miR: Argonaute bound to miRNA. RISC: RNA-induced silencing complex.

**Figure 2 pharmaceuticals-16-01515-f002:**
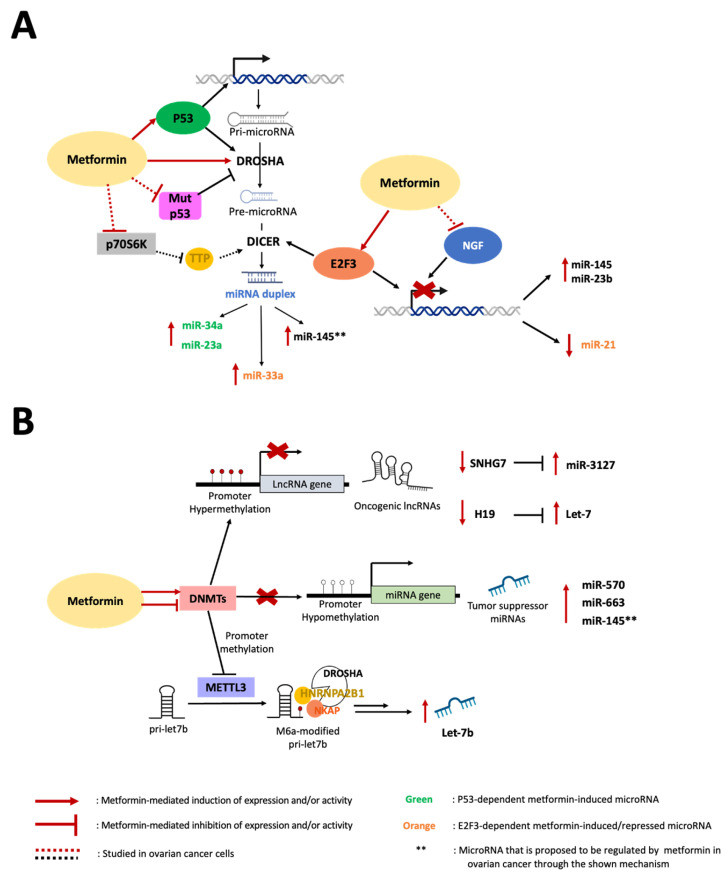
Mechanisms of ncRNA biosynthesis by metformin. (**A**) Metformin induces the expression and activity of wild-type p53 to upregulate miR-34a and miR-23a in carcinoma cells and represses miR-21 expression by promoting the activity of E2F3 on the miR-21 promoter while inducing miR-33a by E2F3-mediated upregulation of DICER. In EOC cells, metformin prevents the NGF-induced decrease in miR-145 and miR-23b and increases the expression levels of these miRNAs. We propose that miR-145 is upregulated by metformin in EOC cells through the induction of p53 (which regulates transcription and maturation), inhibition of mutant p53 expression (which disfavors Drosha activity on pri-miR-145), and Akt/mTOR/p70S6K signaling impairment (which hinders pre-miR-145 processing by Dicer). (**B**) In EOC cells, metformin treatment induces hypermethylation in the H19 and SNHG7 promoters, increasing the bioavailability of let-7 and miR-3127 by releasing them from lncRNA sequestration. Metformin-mediated SNHG7 promoter hypermethylation has been associated with DNMT induction in hypopharyngeal cancer cells. In contrast, metformin treatment produces miR-663 and miR-570 promoter hypomethylation (upregulating their expression) in pancreatic and osteosarcoma cancer cells, respectively. We propose that miR-145 is upregulated by metformin through hypomethylation of its promoter in EOC cells. Metformin inhibits DNMTs binding to the METTL3 promoter, lowering its methylation status, and increases METTL3 expression, allowing m6A modification of pri-let-7b, which binds m6A reader factors (HNRNPA2B1 and NKAP). These produce Dicer recognition and processing of pri-let-7b into mature let-7b.

**Table 1 pharmaceuticals-16-01515-t001:** Effects of metformin on lncRNA levels and its biological effects in different models of cancer. SAHH: S-adenosyl homocysteine hydrolase. d: day. n/a = not assessed in the study.

Cancer	Experimental Model	Metformin Dose	lncRNA	Regulation	Target	Biological Effects	Ref.
Bladder	5637 cells	10 and 20 mM	UCA1	Down	n/a	Inhibits cell proliferation and glycolysis	[113]
Breast	MCF-7 cells	2 mM	H19	Down	SAHH	Reduces cell viability	[82]
	MDA-MB-231 cells	10 and 20 mM	HOTAIR	Down	n/a	Decreases cell viability, migration, and invasion, and suppresses EMT	[114]
	MCF-7 cells	1, 5, and 20 mM	MALAT1	Up	n/a	Inhibits cell proliferation and migration, induces apoptosis, and induces autophagy and ER stress markers	[115]
		1 and 20 mM	HOTAIR				
		20 mM	DICER1-AS1				
		10 and 20 mM	LINCO01121				
		20 mM	TUG1				
	MCF-7 cells	2, 5, and 10 mM	H19	Down	n/a	Induces ferroptosis by inhibiting autophagy	[116]
	MCF-7 cells resistant to tamoxifen	5 mM	GAS5	Up	n/a	Increases sensitivity to tamoxifen by inhibiting cell growth and inducing apoptosis	[117]
Cervical	HCC-94 cells	100 uM	H19	Down	n/a	Inhibits cell viability	[118]
			FTX				
Colorectal	SW480 and SW620 cells	20, 40, and 80 mM	UCA1	Down	n/a	Suppresses cell proliferation and promotes apoptosis	[119]
Endometrial	ARK2 cells and endometrial cancer patients	2 mM in cells, 750 mg/d up to 2250 mg/d in patients	H19	Down	SAHH	Alters DNA methylation genome widely. Reduces cell viability and tumor cell proliferation	[82]
	Endometrial cancer patients	750 mg/d up to 2250 mg/d	H19	Down	n/a	Reduces H19 expression	[120]
Gastric	AGS cells	20 mM	H19	Down	n/a	Inhibits cell proliferation, invasion, and migration, and suppresses metastasis	[121]
	HR, AZ-521, NCI-N87, and TSGH cells	10 mM	H19	Down	n/a	Inhibits cell proliferation and invasion	[122]
		1, 5, and 10 mM	RBMS3-AS3		n/a		
Hypopharyngeal	FaDu cells and xenograft mouse model	2,4,6,8, and 10 mM in cells and 8 mM in mice	SNHG7	Down	SAHH	Inhibits cell proliferation and tumor growth. Sensitizes to taxol and radiotherapy	[74]
Liver cancer	HepG2, SNU-449, and SK-Hep-1 cells	10 and 20 mM	HULC	Down	n/a	Decreases cell growth	[123]
	HepG2 cells	7.57 μg/ml	AF085935	Down	n/a	Inhibits cell proliferation	[124]

**Table 2 pharmaceuticals-16-01515-t002:** Summary of findings obtained by studies that tested the pre-operative use of metformin before cytoreduction or in pre-malignant lesions. IGF-1: insulin-like growth factor 1. IGFBP-7: insulin-like growth factor-binding protein 7. mTOR: mechanistic target of rapamycin.

Pathology	Findings in the Metformin-Treated Group	Ref.
Adenoma and polyp recurrence in patients with a high risk of adenoma recurrence	Reduced the prevalence and number of metachronous adenomas or polyps after polypectomy	[145]
Oral pre-malignant lesions	Decrease in cell proliferation in the squamous epithelium and inhibition of mTOR signaling	[146]
Endometrial cancer	Decrease in Ki-67 and pS6 staining. Decrease in plasma IGF-1 and IGFBP-7	[147]
Newly diagnosed women with breast cancer	Reduced expression of p53, BRCA1, and cell cycle pathways following metformin treatment	[148]
Localized prostatic cancer	Metformin reduced the Ki-67 proliferation index and trended toward prostatic-specific antigen reduction	[149]
Endometrial cancer	Decreased phosphorylated extracellular signal-regulated kinase (ERK1/2), cyclin D1, Ki-67, and topoisomerase IIα. Increased p27	[150]

**Table 3 pharmaceuticals-16-01515-t003:** Clinical trials registered in ClinicalTrials.gov that tested metformin and considered measuring miRNAs.

Study	Patient	Treatment	Aim of Investigation	miRNA
NCT05468554 Thyroid cancer (Phase 3) [151]	Women of reproductive age diagnosed with thyroid carcinoma	Metformin 500 mg, 3 times a day	Evaluate the metformin effect on the fertility of women treated with 131I for thyroid cancer	Difference in the expression of selected miRNAs (uninformed)
NCT03685409 Oral cancer (Phase 3) [152]	Both genders (20–70 years), clinically diagnosed and histologically confirmed as having potential oral malignant lesions	Metformin 500 mg, twice a day	Effect of systemic metformin hydrochloride on the millimeter change in the largest diameter of the potential oral malignant lesion	Numerical differences between miR-21 and miR-200 in tissue biopsies versus saliva at baseline and at 3 months
NCT03684707 Oral cancer (Phase 4) [153]	Both genders (20–70 years), clinically diagnosed and histologically confirmed as having potential oral malignant lesions	Metformin 500 mg, daily	Evaluate lesion size in millimeters	Measured miRNA31 and 210 in saliva and tissue biopsy
NCT05292573 Endometrial cancer (Phase 3) [154]	Women aged ≥20 years with histological diagnosis of simple hyperplasia/complex hyperplasia (SH/CH)	Metformin 500 mg, twice a day	Longitudinal follow-up in women with endometrial hyperplasia without atypia	The area under the receiver operating characteristic curve (ROC curve) (AUC) of the miRNA panel

## Data Availability

Data is contained within the article.

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
