# Peer review of "Mechanisms of Regulation of the Expression of miRNAs and lncRNAs by Metformin in Ovarian Cancer"

_pharmaceuticals, 2023, doi:10.3390/ph16111515_

Round 1

Reviewer 1 Report

The authors focused on Mechanisms of regulation of the expression of miRNAs and lncRNAs by metformin in ovarian cancer. For a review type paper, the manuscript is too poor (9 pages of real manuscript on MDPI format which occupy 2/3 of a page). Extensive revision and improvement must be done before any consideration of this manuscript. Please see below my suggestions:

1.     Lines of the manuscript should have been numbered to facilitate revision, as the MDPI draft provides for each journal.

2.     Initials of each author must be added after each email (in affiliation).

3.     2nd paragraph of Introduction. As you are mentioning about current treatment, a paragraph must be also added about plant-based compounds / traditional medicine, which are in increasing trend, being used more and more in cancer therapeutical approach. I suggest checking and referring to: https://doi.org/10.3390/molecules28020750,https://doi.org/10.1007/s11356-020-09028-0  https://doi.org/10.3390/cancers14194884

4.     Develop the Aim of the study, which is missing. At the final of Introduction, AIM of the study can be better presented. It must be clearly stated and addressed from the perspective of describing the contribution to the field under analysis and the elements of scientific novelty presented, as the LAST, SEPARATE paragraph of this section, to be easier visible. Develop it as better as you can. What differentiate your paper from other in the same topic? Give a reason for interest in this paper.

5.     Please describe in a new, separate 2nd section the Methodological Approaches. How did you have selected the included references? Based on which inclusion/exclusion criteria? Searching which data bases? Published in what period (between which years)? Please detail. Also in this new section, if the authors will make a short search (graphically) related to the impact of the topic on the general literature (using i.e. Web of Science or other comprehensive data vase the authors will consider) they can provide a scientometric figure/Treemap chart (the authors will decide which is more suitable/relevant) which would be interesting in justifying the novelty/impact of the topic in the frame of the existing literature. Do not forget renumbering the following sections.

6.     Title of a Table must be above it, not under.

7.     Acronyms/Abbreviations/Initialisms should be defined the first time they appear in each of three sections: the Abstract; the Main text; the first figure or table. When defined for the first time, the acronym/abbreviation/initialism should be added in parentheses after the written-out form. Apply to the main text and after the title of each figure or under each table where abbreviations have been used on the figure/in the table (explaining all abbreviations used, as many are missing).

8.     A separate subsection is needed about: How is implicated in the mechanisms and how can be evaluated the association between chronic inflammation and cancer, in the case of ovarian cancer? check https://doi.org/10.1016/j.biopha.2023.115015

9.     A new chapter should be done where you detail the impact of metformin on glucose metabolism in the cell and on the rate of cell mitosis. In this new section, make more references towards metformin impact on certain genes related to cell replications such as p53 gene or Ki-67 proliferation index. Also discuss the impact of metformin on tumour angiogenesis.

10.  Title of actual section 3 should be “Conclusions”, as no other type of conclusions are provided.

Good English, minor editing.

Author Response

Reviewer 1

The authors focused on Mechanisms of regulation of the expression of miRNAs and lncRNAs by metformin in ovarian cancer. For a review type paper, the manuscript is too poor (9 pages of real manuscript on MDPI format which occupy 2/3 of a page). Extensive revision and improvement must be done before any consideration of this manuscript. Please see below my suggestions:

  1. Lines of the manuscript should have been numbered to facilitate revision, as the MDPI draft provides for each journal.

Response: First of all, we thank the reviewer for the constructive comments. We used an MDPI draft for this new version, that includes the number of pages and lines.

  1. Initials of each author must be added after each email (in affiliation).

Response: Authors’ initials were added.

  1. 2ndparagraph of Introduction. As you are mentioning about current treatment, a paragraph must be also added about plant-based compounds / traditional medicine, which are in increasing trend, being used more and more in cancer therapeutical approach. I suggest checking and referring to: https://doi.org/10.3390/molecules28020750,https://doi.org/10.1007/s11356-020-09028-0  https://doi.org/10.3390/cancers14194884 

Response:  We added this information. Please, see the second paragraph in the introduction section (highlighted in yellow).

  1. Develop the Aim of the study, which is missing. At the final of Introduction, AIM of the study can be better presented. It must be clearly stated and addressed from the perspective of describing the contribution to the field under analysis and the elements of scientific novelty presented, as the LAST, SEPARATE paragraph of this section, to be easier visible. Develop it as better as you can. What differentiate your paper from other in the same topic? Give a reason for interest in this paper.

Response:  We added the information requested. Please, see the new paragraphs in the introduction section, highlighted in yellow.

  1. Please describe in a new, separate 2ndsection the Methodological Approaches. How did you have selected the included references? Based on which inclusion/exclusion criteria? Searching which data bases? Published in what period (between which years)? Please detail. Also in this new section, if the authors will make a short search (graphically) related to the impact of the topic on the general literature (using i.e. Web of Science or other comprehensive data vase the authors will consider) they can provide a scientometric figure/Treemap chart (the authors will decide which is more suitable/relevant) which would be interesting in justifying the novelty/impact of the topic in the frame of the existing literature. Do not forget renumbering the following sections.

Response: This section was added. Please, see the line 124.

  1. Title of a Table must be above it, not under.

Response:  It was corrected.

  1. Acronyms/Abbreviations/Initialismsshould be defined the first time they appear in each of three sections: the Abstract; the Main text; the first figure or table. When defined for the first time, the acronym/abbreviation/initialism should be added in parentheses after the written-out form. Apply to the main text and after the title of each figure or under each table where abbreviations have been used on the figure/in the table (explaining all abbreviations used, as many are missing).

Response:  Thank you for these corrections. We reviewed and defined the missing abbreviations.

  1. A separate subsection is needed about: How is implicated in the mechanisms and how can be evaluated the association between chronic inflammation and cancer, in the case of ovarian cancer? check https://doi.org/10.1016/j.biopha.2023.115015

Response:  Thank you for the suggestion. We added a paragraph in the new version of the manuscript. Please, check the introduction section.

  1. A new chapter should be done where you detail the impact of metformin on glucose metabolism in the cell and on the rate of cell mitosis. In this new section, make more references towards metformin impact on certain genes related to cell replications such as p53 gene or Ki-67 proliferation index. Also discuss the impact of metformin on tumour angiogenesis.

Response: We added a new section. Please check the second paragraph of the section 1.3. (Metformin in OC).

  1. Title of actual section 3 should be “Conclusions”, as no other type of conclusions are provided.

Response: This change was performed.

Reviewer 2 Report

Dear Authors,

my comments:

1. All abbreviations should be described.

2. What is Argonaute protein?- Should be explained widely

3. epigenetic modifications- should be described widely.

4. In am not sure if Table No 1. is needed.

Author Response

Reviewer 2

Dear Authors,

my comments:

  1. All abbreviations should be described.

Response: First of all, we thank the reviewer for the constructive comments. We reviewed and defined the missing abbreviations in the new version of the manuscript.

  1. What is Argonaute protein?- Should be explained widely

Response: An explanation was added. Please see line 162 in the new version of the manuscript.

  1. epigenetic modifications- should be described widely.

Response: The information was added. Please see line 250 in the new version of the manuscript.

  1. In am not sure if Table No 1. is needed.

Response: Table 1 was removed from the new version.

Reviewer 3 Report

In the present review, the authors discuss the role of metformin-dependent miRNA and lncRNA in ovarian cancer. I have several reservations, my comments are appended below:

1.       Discuss how metformin regulates metastasis through lnc and miRNA.

2.       Authors should also explain the role of metformin in sensitizing to traditional chemo and radiotherapy in solid tumors and also shed light on the involvement of lnc and miRNA>

3.       Discuss and note the clinical trials on the use of metformin in preventing ovarian cancer metastasis and therapy resistance.

4.       Mechanistic details on the Role of transcriptional factors in the metformin-mediated modulation of miRNA expression: use the figure to explain the mechanisms.

5.       Reference 30- authors should note the type of cancer. All similar references should be checked throughout the manuscript.

6.       Authors should tabulate studies documenting the use of metformin in patients with cancer, especially in connection with lnc and miRNAs.

7.       There should be future directions section. 

N/A

Author Response

Reviewer 3

  1. Discuss how metformin regulates metastasis through lnc and miRNA.
  2. Authors should also explain the role of metformin in sensitizing to traditional chemo and radiotherapy in solid tumors and also shed light on the involvement of lnc and miRNA

Response: First of all, we thank the reviewer for the constructive comments. We added this information in different sections in the new version of the manuscript. Please see all the changes highlighted in yellow.

  1. Discuss and note the clinical trials on the use of metformin in preventing ovarian cancer metastasis and therapy resistance.

Response: We added a new section with the suggested changes. Please see the section 6 “Future directions”

  1. Mechanistic details on the Role of transcriptional factors in the metformin-mediated modulation of miRNA expression: use the figure to explain the mechanisms.

Response: We added the figure requested. Please check Figure 2 “Mechanisms of ncRNA biosynthesis by metformin”

  1. Reference 30- authors should note the type of cancer. All similar references should be checked throughout the manuscript.

Response: We checked the models and references and added or corrected any errors and omissions. Please see the new information highlighted in yellow in the new version of the manuscript.

  1. Authors should tabulate studies documenting the use of metformin in patients with cancer, especially in connection with lnc and miRNAs.
  2. There should be future directions section. 

Response: All the suggested changes were included in the new section 6 “Future directions”

Reviewer 4 Report

This review is a narrative review that focused on mechanisms of regulation of the expression of miRNAs and lncRNAs by metformin in ovarian cancer. This review is interesting and I have only minor comments.

Please cite the latest statistics data.

Reference [1] should be revised.

Introduction

Despite optimal surgery and appropriate first-line chemotherapy, 70%–80% of patients with EOC will develop disease relapse after months of therapy[4,5], indicating that it is necessary to improve the existing therapies.

The authors need to revise this paragraph. There is a possibility of the facts being misunderstood.

Table 1

This table is hard to see.

Please delete the highlighted yellow.

What are FuOX resistant cells?

I recommend that the manuscript be reviewed by a person with professional proficiency in English to correct errors in grammar, punctuation, word choice, and sentence construction to improve the flow of ideas expressed in the article to ensure that the document reads as though written by a native English speaker.

Author Response

Reviewer 4

This review is a narrative review that focused on mechanisms of regulation of the expression of miRNAs and lncRNAs by metformin in ovarian cancer. This review is interesting and I have only minor comments.

Please cite the latest statistics data.

Reference [1] should be revised.

 Response: We thank the reviewer for their constructive comments. We updated the reference 1.

Introduction. Despite optimal surgery and appropriate first-line chemotherapy, ∼70%–80% of patients with EOC will develop disease relapse after months of therapy[4,5], indicating that it is necessary to improve the existing therapies.

The authors need to revise this paragraph. There is a possibility of the facts being misunderstood.

 Response: We agree with this comment. The paragraph was corrected.

Table 1

This table is hard to see.

Please delete the highlighted yellow.

What are FuOX resistant cells?

Response: Table 1 was removed from the manuscript, according to the suggestions of other reviewer.

Comments on the Quality of English Language

I recommend that the manuscript be reviewed by a person with professional proficiency in English to correct errors in grammar, punctuation, word choice, and sentence construction to improve the flow of ideas expressed in the article to ensure that the document reads as though written by a native English speaker.

Response: The manuscript was reviewed by a native English speaker. Please check the changes highlighted in yellow.

Reviewer 5 Report

The authors conducted a well-organized review of what has been published to date about the mechanisms of action of metformin in the regulation of miRNAs and lncRNAs with particular attention to what happens in ovarian cancer cells.

The review is well updated and detailed and is the first to draw attention to this topic.

I only have a few points to highlight:

1. In the caption of Figure 1 I see written “Pri-miR: preimary miR” when it should be “primary miRNA” and also “Pre-miR: precursor miR” when it should be “Pre-miR: Precursor miRNA”

2. In Figure 1 – “exporin 5” should be “exportin 5”

3. Please check typographical errors throughout the text.

Author Response

Revisor 5

The authors conducted a well-organized review of what has been published to date about the mechanisms of action of metformin in the regulation of miRNAs and lncRNAs with particular attention to what happens in ovarian cancer cells.

The review is well updated and detailed and is the first to draw attention to this topic.

I only have a few points to highlight:

  1. In the caption of Figure 1 I see written “Pri-miR: primary miR” when it should be “primary miRNA” and also “Pre-miR: precursor miR” when it should be “Pre-miR: Precursor miRNA”

Response: We thank the reviewer for the constructive comments. We performed the changes suggested in Figure 1 in the new version of the manuscript.

  1. In Figure 1 – “exporin 5” should be “exportin 5”

Response: We corrected this mistake.

  1. Please check typographical errors throughout the text.

Response: We checked for typos and performed several changes (please, see the changes highlighted in yellow). In addition, the manuscript was reviewed by a native English speaker.

Round 2

Reviewer 1 Report

The authors improved their manuscript

Minor errors.

Reviewer 2 Report

Dear Authors,

I accept your reply

Reviewer 3 Report

All my concerns were addressed. 

I recommend to accept